# Pre-Dialysis B-Line Quantification at Lung Ultrasound Is a Useful Method for Evaluating the Dry Weight and Predicting the Risk of Intradialytic Hypotension

**DOI:** 10.3390/diagnostics12122990

**Published:** 2022-11-29

**Authors:** Marco Allinovi, Giulia Palazzini, Gianmarco Lugli, Iacopo Gianassi, Lorenzo Dallari, Selene Laudicina, Marco Gregori, Francesco Rossi, Daniele Giannerini, Roberta Cutruzzulà, Egrina Dervishi, Maria Biagini, Calogero Lino Cirami

**Affiliations:** 1Nephrology, Dialysis and Transplantation Unit, Careggi University Hospital, Largo Brambilla, 3, 50134 Florence, Italy; 2Department of Biomedical Experimental and Clinical Sciences “Mario Serio”, University of Florence, 50121 Florence, Italy; 3Nephrology Unit, Meyer Children’s Hospital, 50139 Florence, Italy; 4Department of Nephrology, Transplantation, Dialysis and Apheresis, Pellegrin University Hospital, 33076 Bordeaux, France

**Keywords:** lung ultrasound, B-lines, dry weight, intradialytic hypotension, fluid overload

## Abstract

Intradialytic hypotension (IDH) is a frequent and well-known complication of hemodialysis, occurring in about one third of patients. An integrated approach with different methods is needed to minimize IDH episodes and their complications. In this prospective observational study, recruited patients underwent a multiparametric evaluation of fluid status through a lung ultrasound (LUS) with the quantification of B-lines, a physical examination, blood pressure, NT-proBNP and chest X-rays. The evaluation took place immediately before and at the end of the dialysis session, and the patients were divided into IDH and no-IDH groups. We recruited a total of 107 patients. A pre-dialysis B-line number ≥ 15 showed a high sensitivity in fluid overload diagnosis (94.5%), even higher than a chest X-ray (78%) or physical examination (72%) alone. The identification at the beginning of dialysis of <8 B-lines in the overall cohort or <20 B-lines in patients with NYHA 3–4 class are optimal thresholds for identifying those patients at higher risk of experiencing an IDH episode. In the multivariable analysis, the NYHA class, a low pre-dialysis systolic BP and a low pre-dialysis B-line number were independent risk factors for IDH. At the beginning of dialysis, the B-line quantification at LUS is a valuable and reliable method for evaluating fluid status and predicting IDH episodes. A post-dialysis B-line number <5 may allow for an understanding of whether the IDH episode was caused by dehydration, probably due to due to an overestimation of the dry weight.

## 1. Introduction

Intradialytic hypotension (IDH) is a serious, well-known complication of hemodialysis (HD). The National Kidney Foundation’s Kidney Disease Outcomes Quality Initiative (KDOQI) guidelines define IDH as a decrease in either systolic BP ≥20 mmHg or mean arterial pressure ≥10 mmHg associated with symptoms or need for intervention [1]. IDH and other hypovolemia-related symptoms are common in HD, occurring in 15% to 50% of ambulatory HD sessions [2,3]. This high range in prevalence reported is due to the different hypotension definitions used. When symptom-based definitions of hypotension are used, the IDH is underestimated because these symptoms (such as dizziness, nausea, headache and muscle cramp) are frequently not reported by patients, but even asymptomatic IDH is a predictor of mortality and non-fatal cardiovascular disease [4,5]. IDH occurs owing to an imbalance between the ultrafiltration rate and the normal compensatory mechanisms, including the plasma refilling rate and a reduction in venous capacity [6]. Moreover, structural and functional abnormalities of the heart and blood vessels (such as those due to a dysautonomic neuropathy, diabetes, being elderly or heart failure) increase the sensitivity of the patient to changes in fluid status [1,2,3,4,5]. Regardless of the definition, IDH has been associated with subsequent vascular access thrombosis, inadequate dialysis dose, cardiac arrhythmias, major adverse cardiac events, dementia, and mortality [3,7,8,9,10,11,12,13,14]. Even when asymptomatic, IDH is associated with a lower white matter and hippocampus volume and reduced global cognitive functioning [15]. Therefore, any IDH episode should prompt a reassessment of fluid status and dry weight estimation, including an evaluation of the UF rate, dialysis treatment time, interdialytic weight gain, and antihypertensive medication use. Every strategy used in hypotension management must be tightly balanced in light of the risk of generating secondary opposite side effects and complications such as fluid overload, interdialytic hypertension, edema, congestive heart failure, inadequate blood purification and the discontinuation of antihypertensive medication. Therefore, an integrated approach to IDH management is needed to resolve acute hypotension crises and to minimize complications. From this perspective, a personalized approach should be applied with a global view to ensuring a management tailored to the individual’s characteristics.

In this complex setting, a deeper knowledge about the main hypotension risk factors is needed. Critical issues include target weight, residual renal kidney function protection, hemodialysate composition, temperature biofeedback and the hemodialysis technique used, since convective therapies lead to major intradialytic stability [16,17,18,19]. Moreover, given the complexity of the involved factors, the dry weight assessment of dialysis patients remains a challenge and predicting an IDH episode using conventional parameters and models may be difficult. Among several approaches, lung ultrasound (LUS) is increasingly being used at the bedside to integrate the clinical assessment of patients on dialysis [20]. The pre-dialysis number of B-lines showed a direct correlation with interdialytic weight gain and the quantification of B-lines is a widely recognized method for assessing the dry weight, decreasing ambulatory blood pressure (BP) values, recurrent episodes of decompensated heart failure and cardiovascular events of patients on dialysis [21,22,23,24,25]. However, comprehensive data regarding the potential role of LUS in predicting the risk of IDH are lacking. In two different randomized clinical trials, an LUS-guided strategy for dry weight assessment reported conflicting results: the first showed a marginal (almost significant) decrease in the percentage of patients experiencing one or more IDH episodes (34.3% vs. 55.6%, *p* = 0.072), while the other showed a 26% higher relative risk of intradialytic cramps (HR 1.26) [23,26]. 

This prospective cohort study aimed to evaluate the potential role of LUS, in particular that of B-line quantification, in predicting IDH episodes and detecting patients at a greater risk of developing IDH because of their fluid status or an overestimation of their dry weight. We also evaluated the relationship between IDH episodes and different parameters to help clarify the most important predictive factors of IDH.

## 2. Materials and Methods

### 2.1. Study Cohort

This was a prospective observational single-center study performed between 1 January 2020 and 1 January 2021 at Careggi University Hospital, a tertiary hospital facility in Florence, Italy. The primary outcome was the development of an IDH episode, defined as a decrease in systolic BP of ≥20 mmHg and/or a decrease in mean arterial pressure of ≥10 mmHg from pre-dialysis levels [1], associated with intradialytic or post-dialytic symptoms (abdominal discomfort, yawning, sighing, nausea, vomiting, muscle cramps, restlessness, dizziness or fainting and anxiety), a need for nursing interventions and/or a failure to meet a prescribed ultrafiltration goal. In relation to the achievement of the primary outcome (an IDH episode) experienced on the same day as the multiparametric evaluation, patients were divided into a hypotension group (IDH) and non-hypotension group (no-IDH). Each patient underwent a single multiparametric evaluation of their fluid status immediately before and at the end of a midweek dialysis session, using the following methods: LUS, physical examination, blood pressure measurement, NT-proBNP and chest X-rays. At the time of patient recruitment, before the hemodialysis session, it was not possible to predict which group the patient would be included in, and in order not to create selection bias, all patients of our Dialysis Center were progressively recruited. No patient refused to complete the pre-dialysis and post-dialysis multiparametric evaluation of fluid status. Inclusion criteria were as follows: (1) age > 18 years; (2) treatment by intermittent hemodialysis, performed 3 times a week; (3) a history of hemodialysis of more than 3 months; and (4) the ability to provide consent. Exclusion criteria were as follows: (1) A pre-dialysis systolic BP < 90 mmHg. (2) A history of pneumonia in the previous weeks, co-existent lung fibrosis or interstitial lung disease, which are diseases that appear as multiple B-lines during an LUS regardless of the fluid status. In particular, patients affected by COVID-19 were excluded for 6 months after infection due to the potential persistence of B-lines [27]. (3) Insufficient clinical information and no, or lacking, laboratory findings. Demographic data and medical history were extracted from the participants’ clinical records. Laboratory parameters and any other dialysis data, such as the UF rate and dialysis prescription, were collected. Data were immediately recorded on individual case report forms. The pre-dialysis systolic BP was measured in a seated position after 5 min of rest, and either every 60 min or in the case of the onset of symptoms. Data on left ventricular hypertrophy (LVH) were derived from our institution’s periodic echocardiography, which is performed every year. Dry weight was assessed according to clinical criteria (blood pressure, presence of peripheral edema or pulmonary crackles, previous IDH episodes and cramps) which represent the gold standard (“trial and error”) approach. The nephrologist in charge of the dialysis service performed a physical examination for the fluid status assessment. The prescribed UF volume was determined based on the patient’s medical history, physical examination and the difference between the pre-dialysis weight and dry weight. The nephrologists who performed LUS were blinded to the health status of the subjects, their interdialytic weight gain, laboratory values and physical examination. Intermittent hemodialysis was performed with 5008 (Fresenius Medical Care, Waltham, MA, USA) or Artis (Baxter Healthcare Corp., Deerfield, IL, USA) hemodialysis machines, and dialysate concentrate solutions with a 1.5 mmol/L calcium concentration. Symptomatic IDH episodes, as well as any other disorder, were recorded during dialysis sessions, together with the administration of fluids or drugs.

### 2.2. Ultrasound Evaluation

LUS was chosen because it is a non-invasive, radiation-free and inexpensive technique, it is easy to learn and interpret and can be performed at the bedside with a portable device and does not depend on acoustic windows or patient position [28]. Although the potential harmlessness of LUS was questioned in preclinical studies, LUS is nowadays widely validated, recommended by the guidelines and its role in reducing numerous strong outcomes has been demonstrated, with no safety issues raised so far in humans. Ultrasound examinations were performed at the bedside immediately before and after dialysis sessions using an ultrasound machine (MyLab Class C-Esaote^®^, Genoa, Italy) with a 6–18 MHz linear probe (LA435). Various transducers (convex, microconvex, linear and phased array) have been used to quantify B-lines in adult patients. We used the linear one because it is thought to be the best for studying the pleural line. A single B-line appears as a hyperechoic, laser-like, vertical line originating from the pleura and extending to the bottom of the field of view, moving in synchrony with the patient’s breathing. There are several different B-line shapes (e.g., a single thin line, a single comet-shaped line, a double or triple line converging to a single point on the pleural line), but all of them should be considered as a single B-line. B-line identification and quantification were performed by physicians with long-term expertise in LUS. A standardized 28-position B-line score was adopted to calculate the cumulative number of B-lines as an expression of interstitial pulmonary congestion, with the patient in the semi-supine position [29]. This approach consists of scanning from the second to fifth intercostal spaces on the right side and from the second to fourth spaces on the left side, in the parasternal, midclavicular, anterior axillary and midaxillary positions, for a total of 28 positions. At every scanning site in each intercostal space, B-lines were counted from 0 to 10. It is currently accepted that, with a 28-position LUS score, a B-line score ≤ 5 is indicative of euvolemia and a B-line score > 15 reflects hypervolemia [30].

### 2.3. Chest X-ray

A chest radiograph, obtained in the orthostatic posteroanterior and lateral projections, gives valuable information for fluid assessment. The presence of enlargement of the azygos vein, enlargement and loss of definition of hilar structures, septal lines in the lower lung (namely, Kerley A- and B-lines), peribronchial and perivascular cuffing with widening and blurring of the margins, thickening of interlobar fissures with subpleural fluid accumulation and/or pleural effusion during chest X-ray imaging was considered as suggestive of lung congestion and hypervolemia [31,32]. Each patient underwent a chest X-ray before the dialysis session.

### 2.4. Blood Volume Monitoring

Blood volume monitoring is routinely performed in all patients undergoing HD in our unit. Nowadays, most manufacturers have incorporated a relative blood volume (RBV) monitor in their dialysis apparatus in order to monitor the RBV slope. In our cohort, the blood volume change during dialysis was monitored using a HEMOcontrol BV sensor (Baxter Healthcare Corp., Deerfield, IL, USA) or Blood Volume Management (BVM^®^) technology (Fresenius Medical Care, Concord, Hopkins, MN, USA). During ultrafiltration, as fluid is removed from an hemodialysis patient’s vascular space, the RBV slope (%) continuously correlates with the increase of hematocrit or total proteins. Trained dialysis nurses recorded the BV reduction every hour during the session.

### 2.5. Statistical Analysis

Parametric data were reported as the mean ± standard deviation (SD) and non-parametric data as the median and interquartile range (IQR) from the 25th to 75th percentile. Continuous variables were compared using a non-parametric Mann–Whitney test, while proportions were compared using a Chi-square test. Multiple variables were compared between groups of sessions (with IDH vs. no-IDH). The IDH was considered a dummy variable, where 0 means “no IDH” and 1 means “presence of IDH”. As appropriate, relationships between variables were tested with the Pearson product-moment correlation coefficient, or the Spearman’s rank correlation coefficient. Logistic regression with both categorical and continuous independent variables was used to build predictive models for the occurrence of hypotension. Univariate receiver operating characteristic (ROC) curve analysis was used to calculate the area under the curve (AUC) with a 95% confidence interval (CI). The maximum value of Youden’s index was applied to ROC curves as a criterion to select the optimum cut-off point both for sensitivity and specificity. A *p*-value < 0.05 was considered statistically significant. Statistical analysis was performed using the SPSS 22.0 software package (IBM, Armonk, NY, USA).

## 3. Results

### 3.1. Baseline Characteristics

From January 2020 to January 2021, a total of 107 patients on chronic hemodialysis received a single multiparametric evaluation characterized by a physical examination, LUS, chest X-ray, NT-proBNP and other laboratory tests. The duration of ultrasound bedside assessments ranged from 5 to 8 min. The main demographic, anthropometric, clinical, biochemical and ultrasound characteristics of the three patient groups (all patients, IDH and no-IDH) are detailed in Table 1. The IDH and no-IDH groups were not significantly different in terms of the prevalence of diabetes, left ventricular hypertrophy (LVH), age and dialysis vintage. Although two of the most important determinants of IDH are the interdialytic weight gain and impaired compensatory mechanisms (especially reduced venous refilling because of hypoalbuminemia), neither an interdialytic weight gain nor hypoalbuminemia resulted in significant differences in the IDH group (Table 1). The IDH group had significantly fewer patients with ≥15 B-lines at pre-dialysis LUS assessment (30.3% vs. 60.8%, *p* = 0.004) and more patients with ≤5 B-lines (42.4% vs. 16.2%, *p* = 0.004), compared to no-IDH patients. Moreover, the IDH group was characterized by a lower pre-dialysis systolic BP (126 mmHg vs. 137 mmHg; *p* = 0.003).

### 3.2. LUS Yields a High Sensitivity in Fluid Overload Diagnosis

In patients with a pre-dialysis B-line score ≥ 15, LUS demonstrated 94.5% sensitivity in fluid overload diagnosis (defined through a clinical evaluation and chest X-ray), while clinical evaluation alone demonstrated 72% sensitivity, and chest X-ray alone demonstrated 78% sensitivity. Patients without B-lines or with a low pre-dialysis B-line score (≤5 B-lines) experienced an IDH episode more commonly (*p* = 0.004).

A binomial logistic regression was performed to ascertain the effects of the pre-dialysis B-line score on the likelihood that participants presented fluid overload (Figure 1B). The logistic regression was statistically significant (χ^2^ = 114.344, *p* < 0.001). An increase in the B-lines score of one conferred 1693-times higher odds of exhibiting fluid overload (95% CI 1.328–2.159, *p* < 0.001). ROC analysis was conducted, and the area under the ROC curve was 0.983 (95% CI 0.964–1.000), which was an outstanding level of discrimination according to Hosmer et al. [33].

### 3.3. The Number of B-Lines Predicts the Risk of IDH

A binomial logistic regression was performed to ascertain the effects of the pre-dialysis number of B-lines on the likelihood that participants had an IDH episode. The logistic regression model was statistically significant (χ^2^ = 15.98, *p* < 0.001). An increase in the number of B-lines of one conferred a 0.946-times lower odds of experiencing an IDH episode (95% CI 0.913–0.979, *p* = 0.003), indicating that a higher B-line score was associated with a decreased likelihood of IDH. ROC analysis was conducted, and the area under the ROC curve was 0.736 (95% CI 0.637–0.835), which was an acceptable level of discrimination according to Hosmer et al. [33] (Figure 1A). According to the Youden index method, the optimal threshold was reached when eight B-lines were detected, with a sensitivity of 63.6% and a specificity of 75.7%. With regard to the post-dialysis number of B-lines, a lower number of B-lines was detectable in patients who experienced IDH (OR 0.895, 95% CI 0.7830–0.9684). ROC analysis retrieved an AUC that showed an acceptable level of discrimination (AUC = 0.734, 95% IC 0.5843–0.8841), with an optimal threshold of five B-lines (Figure 1C).

### 3.4. The Number of B-Lines Predicts the Risk of IDH, Even in Patients with Heart Failure

A subgroup analysis was conducted according to the NYHA class (see Appendix A). Notably, in NYHA classes 0–2, the optimal threshold to predict an IDH episode was reached when 8 B-lines were detected (sensitivity 70.5%, specificity 75%), but in NYHA classes 3–4, a threshold as high as 20 B-lines was reached (sensitivity 92.3%, specificity 84.6%).

In our cohort of 26 patients with heart failure (defined as NYHA class ≥ 3), only a chest X-ray, the NT-proBNP value and the number of B-lines before dialysis appeared to be promising fluid status assessment methods that were able to predict an IDH episode (Table 2).

### 3.5. The Number of B-Lines Correlates with NT-proBNP and Serum Albumin

Correlation analysis was run to assess the relationship between the number of B-lines and the other variables. Preliminary analyses showed that only the weight difference (weight—dry weight) in Kg and weight difference in %, the two variables which describe the interdialytic weight gain, were normally distributed, as assessed using Shapiro–Wilk’s tests (*p* > 0.05). Thus, a Pearson’s product-moment correlation was run to assess the relationship between the number of B-lines and these variables, showing in both cases a positive statistically significant correlation (r(107) = 0.529, *p* < 0.001; r(107) = 0.586, *p* < 0.001). A Spearman’s rank-order correlation was run to assess the relationship between the number of B-lines and the other variables measured. A preliminary analysis showed that the relationship with the NT-proBNP plasma concentration, albuminemia and NYHA class was monotonic, as assessed by a visual inspection of scatter plots. There was a statistically significant, strong positive correlation between the number of B-lines and the NT-proBNP plasma concentration (r_s_(77) = 0.628, *p* < 0.001) and a positive correlation with the NYHA class (r_s_(107) = 0.301, *p* = 0.002) and hypoalbuminemia (r_s_(107) = −0.194, *p* = 0.046). 

### 3.6. Hypoalbuminemia, Low Pre-Dialysis Systolic BP and a Low Number of B-Lines Are Independent Risk Factors for IDH

Table 3 summarizes the results of the multivariable logistic regression analysis of 12 factors in relation to IDH. We decided to include all these variables in this analysis (Model 1) without testing each one in univariable logistic regression models to preserve all the possible clinical information related to IDH. In the multivariable analysis (Table 3, Model 1), a low pre-dialysis systolic BP (OR 0.964; 95% CI 0.933–0.996) and a low pre-dialysis B-line score (OR 0.877; 95% CI 0.817–0.942) were independent risk factors for IDH (*p* for the model < 0.01). Model 2 was built including only independent risk factors derived from Model 1, with the inclusion of NYHA class and hypoalbuminemia, both of which had *p* < 0.1 in Model 1. A low pre-dialysis systolic BP, low number of B-lines, and NYHA class, but not hypoalbuminemia, were independent risk factors for developing IDH (*p* for the model < 0.01).

### 3.7. IDH and Pre-Dialysis B-Lines Are Not Associated with a Higher Mortality

Among patients with at least one year of follow-up, the overall one-year mortality rate was 33.7% and it did not differ between IDH and no-IDH patients (43.8% vs. 28.3%, *p* = 0.136). Moreover, neither the number of pre-dialysis B-lines nor belonging to the IDH group were significant predictors of one-year mortality (binary logistic regression: *p* = 0.554 and *p* = 0.480).

### 3.8. Overall Performance of Selected Items to Predict IDH

We then decided to assess the performance of single variables for the prediction of IDH. We tested LUS (in terms of number of B-lines before dialysis), the BVM values (in terms of % slope in RBV during first hour of dialysis), the clinical evaluation (based on the physical examination), the chest X-rays and the NT-proBNP values in relation to the likelihood that participants had intradialytic hypotension (Figure 2A). In the ROC analysis, the number of B-lines performed significantly better than all the other variables, with the exception of BVM (Figure 2B). Taken together, these results highlighted the feasibility of using LUS to predict IDH and its superiority to chest X-rays and physical examination, which are widely used.

With regard to RBV changes during the first hour of dialysis, and according to the Youden index method, the optimal diagnostic threshold was reached when a −7.9% reduction in RBV during first hour of dialysis was detected, with a sensitivity of 82.7% and a specificity of 62.5%. With regard to the Nt-proBNP values, and according to the Youden index method, the optimal diagnostic threshold was reached when a Nt-proBNP value of 33713.5 was detected, with a sensitivity of 40.4% and a specificity of 91.7%.

## 4. Discussion

IDH is a frequent complication in hemodialysis and its importance is shown by the important adverse clinical outcomes it determines, both in the short- and long-term. In our study, 33/107 (30.4%) patients on three-weekly hemodialysis experienced an IDH episode. This prevalence was in line with previously published articles, in which IDH was documented in 5–77.7% of patients [34]. This huge range depends on the definition of IDH used, but the majority of the studies confirmed a prevalence between 15 and 30% of all sessions [35]. Several risk factors of IDH have been identified by different studies over the years, such as age, hypoalbuminemia, ultrafiltration rate (mL/h/kg) and pre-dialysis SBP, and all of them were confirmed in our study [3,36,37]. From a recent meta-analysis, diabetes, high interdialytic weight gain, female gender, and lower body weight were described in association with IDH across studies, but we did not confirm these associations in our cohort [34]. We also demonstrated that an NYHA class ≥ 3, a low number of B-lines before dialysis, and a low pre-dialysis systolic BP were independent risk factors for IDH. The use of LUS as a tool for assessing patients’ hydration status has been spreading for some years, and it was established that a finding of ≤5 B-lines was indicative of euvolemia and that >15 B-lines reflected hypervolemia, considering a 28-position LUS score [30]. By applying this approach to dialysis patients, for whom the hydration state is so variable, we demonstrated that LUS is a valuable technique for fluid overload assessment. We found a higher incidence of IDH in patients with fewer pre-dialysis B-lines and a lower risk of IDH in cases where pre-dialysis there was ≥15 B-lines. A similar study showed that there was an increased risk of IDH in critically ill patients on intermittent HD with <14 pre-dialysis B-lines only if they presented a concomitant vena cava collapsibility ≤ 11.5 mm m^−2^ [38]. Another study, again on patients undergoing HD in an intensive care setting, showed that patients with documentation of an A-line pattern (indicating a dry lung) had a higher incidence of IDH than those with an overriding B-line pattern, although the authors did not provide a prognostic threshold in the B-line score [39]. In our study, a significantly lower number of B-lines post-dialysis was identified in patients who experienced IDH, with ≤5 B-lines in most patients, highlighting the central role of hypovolemia in IDH etiopathogenesis. Altogether, we demonstrated that, both pre-dialysis and post-dialysis, the number of B-lines was inversely related to the risk of an IDH episode. The identification of <8 B-lines using LUS at the beginning of dialysis was an optimal threshold for identifying those patients at higher risk of experiencing an IDH episode, with a sensitivity of 63.6% and specificity of 75.7%, and well-designed prospective studies are required to validate this observation. Although these values were not excellent, we should also consider that the pathophysiology of IDH events is very complex and not completely predictable. Not all hypotension is due to hypovolemia, and not all hypovolemia causes significant hypotension. This happens because there are compensatory mechanisms that try to maintain a stable BP. Consequently, LUS before dialysis cannot aspire to predict all IDH episodes and we consider the results obtained as satisfactory. Conversely, a post-dialysis B-lines quantification may provide an understanding of whether the IDH episode was due to dehydration, probably due to an overestimation of the dry weight, as demonstrated by <5 B-lines, or due to other reasons (e.g., autonomic dysfunction, high UF rate, non-dialyzable antihypertensive drugs, hypoalbuminemia and/or anemia).

Even considering the multifactorial nature of IDH in hemodialysis and the great heterogeneity of dialysis patients, a multiparametric approach to predict IDH is needed. Consequently, nephrologists should also be careful about UF rate prescription (in particular for >13 mL/kg/hour), and be more vigilant about RBV changes (in particular for an RBV decline below −7.9% during the first hour of dialysis).

Each method (physical examination, B-line score, blood pressure measurement, NT-proBNP and RBV decline) suffers from several shortcomings, and, consequently, we should adopt different methods or different thresholds for each method in the presence of comorbidities associated with autonomic dysfunction. For example, in patients with severe heart failure (defined as NYHA class ≥ 3), the loss of compensation from increased contractility predisposes them to the development of IDH even with >8 B-lines pre-dialysis or with physical signs of fluid overload. In these patients, pulmonary congestion (identified using a chest X-ray or the number of B-lines) did not appear to be representative of systemic fluid status, and this was likely related to the concomitant left ventricular dysfunction. Interestingly, the identification of <20 B-lines pre-dialysis in patients with NYHA class 3–4 was the optimal threshold for predicting an IDH episode, with a sensitivity of 92.3% and specificity 84.6%, suggesting that the extent of pulmonary congestion in these patients was not representative of the overall volume status. A certain amount of pulmonary congestion remained, even when the patient had reached the dry weight (Table 2). As a result of left ventricular dysfunction, we should be very careful to avoid systemic dehydration (and, therefore, IDH) in an attempt to dry out the lung congestion.

Although the development of IDH is recognized as a risk factor for mortality in dialysis patients, in our study, we only found a correlation between IDH and one-year mortality (Table 1). Statistical significance was probably not reached due to the relatively small number of patients enrolled with an adequate follow-up.

Changes in the B-line score were apparent in real-time with fluid removal during dialysis, reflecting the degree of interstitial imbibition of lung tissue, with a progressive B-line score reduction over the hemodialysis session [40,41]. This versatility of LUS makes it a useful tool, not only at the beginning of dialysis, but also during or at the end of the session, to guide a progressive reduction in post-dialysis weight in order to optimize patients’ baseline fluid status, even considering that IDH generally occurs at a median time interval of 120–149 min [42]. We recommend LUS as a routine exam in patients on dialysis in order to obtain a more accurate volemic status evaluation, but also to prevent IDH episodes. In particular, LUS can be used as a first-level exam before dialysis sessions in patients at a high risk of developing IDH, such as those with (1) frequent or recent IDH episodes, (2) fluctuations in dry weight, (3) recent surgery or infections, (4) malnutrition, (5) a recent period of fasting or an increased appetite and (6) glucocorticoid therapy.

In this study, we also demonstrated that a simple pulmonary assessment using LUS provided relevant information about pulmonary congestion in hemodialysis patients (outperforming chest radiography [43]) and identified patients at risk of complications. This new element can help an integrated clinical evaluation (B-line quantification together with BVM and a physical examination) and guide physicians in the selection of an appropriate ultrafiltration profile. We propose that dialysis units adopt LUS in their daily clinical practice as a bedside tool not only for fluid status assessment and dry weight prescription, but also to prevent IDH and drive the ultrafiltration prescription during the whole hemodialytic session (Figure 3). The risk of an IDH episode can be reduced at the beginning of any dialysis session with treatment of the modifiable risk factors, for example, by administering colloids (serum albumin) during the dialysis session, modulating antihypertensive therapies (for patients with a low pre-dialysis systolic BP) and by assessing the number of B-lines pre-dialysis (as a sensitive method with which to evaluate fluid status).

The small number of patients enrolled from one single center represents an important limitation of this study, and it was not sufficiently powered to detect clinically relevant changes in strong outcomes, such as mortality or cardiovascular events. Due to its observational nature and the lack of a follow-up, this study could not establish whether the routine use of LUS could improve efficiency in terms of preventing IDH episodes, but it does serve as a pilot study for future studies. Last, we recruited consecutive patients who needed a multiparametric evaluation of fluid status and, consequently, we might have selected patients more prone to fluid status abnormalities, including both hypovolemia or hypervolemia. Despite this, patients’ characteristics were similar between the two groups.

## 5. Conclusions

In summary, IDH occurs in response to a reduction in blood volume as an expression of different independent risk factors, such as abnormal cardiac function (NYHA class ≥ 3), lower pre-dialysis systolic BP and lower fluid status (quantified by LUS as a lower number of B-lines). Using LUS at the beginning of dialysis is a valuable method for fluid status assessment, showing a high sensitivity in fluid overload diagnosis, even higher than a chest X-ray and physical examination. The pre-dialysis number of B-lines at LUS assessment was able to predict an IDH episode independently from the NYHA class, UF rate and physical signs/symptoms of fluid status. A low B-line score (<8 B-lines) at the beginning of dialysis may predict IDH and its quantification should be integrated with other clinical and laboratory parameters in order to drive the prescription of ultrafiltration. A low B-line score (≤5 B-lines) at the end of dialysis may suggest that IDH was associated with hypovolemia due to an overestimation of the dry weight. In patients with low B-line scores, nephrologists should be careful about UF prescription and more vigilant about RBV changes during a dialysis session.

## Figures and Tables

**Figure 1 diagnostics-12-02990-f001:**
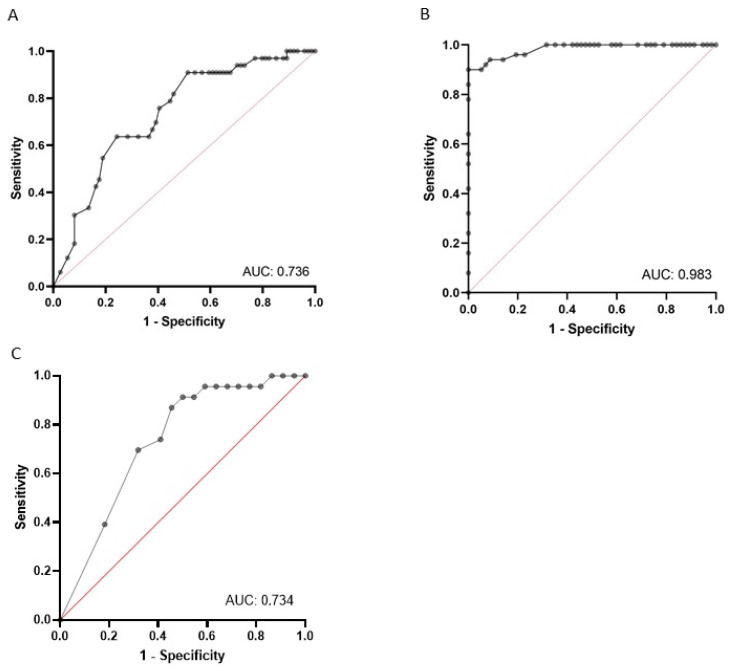
ROC curves in the overall cohort. (**A**) ROC curve plotting number of B-lines with the likelihood that participants had intradialytic hypotension; (**B**) ROC curve plotting number of B-lines with the likelihood that participants had overall fluid overload; (**C**) ROC curve plotting number of post-dialytic B-lines with the likelihood that participants had intradialytic hypotension.

**Figure 2 diagnostics-12-02990-f002:**
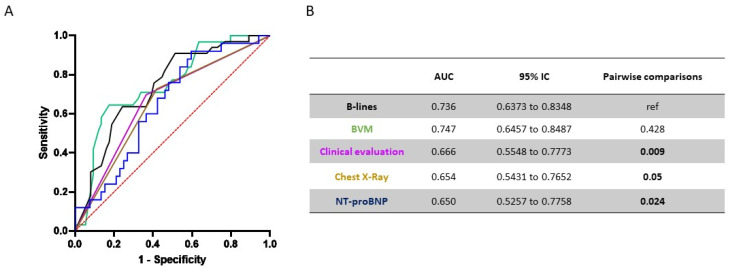
Comparative analysis of different fluid status assessment methods to predict IDH episodes. ROC curve analysis for the overall cohort. (**A**) ROC curves plotting number of B-lines, BVM values, clinical evaluation, chest X-ray and the NT-proBNP values in relation to the likelihood that participants had an IDH. (**B**) Table shows AUC values and pairwise comparisons with B-lines score.

**Figure 3 diagnostics-12-02990-f003:**
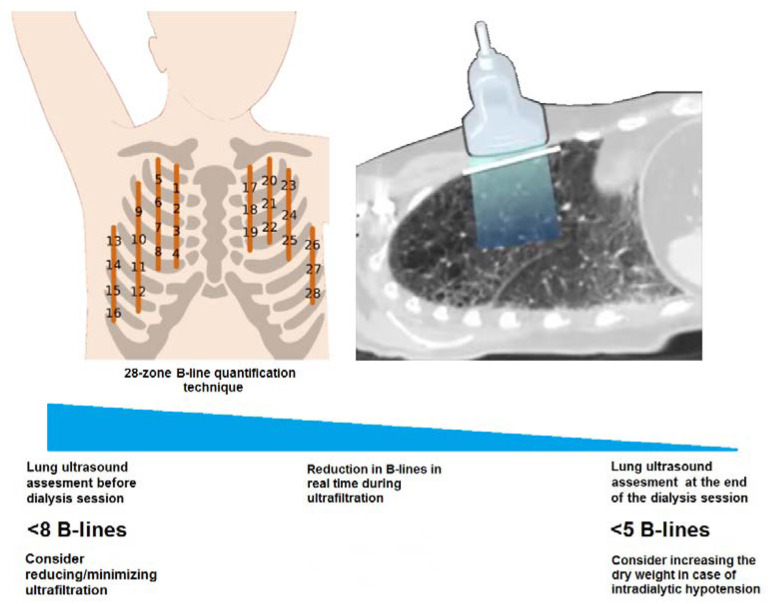
A proposed lung ultrasound approach in hemodialysis in order to drive ultrafiltration prescriptions. Nephrologists should adopt B-line quantification by LUS as a bedside approach to prevent IDH and drive the ultrafiltration prescription during the whole hemodialytic session. The identification of <8 B-lines at the beginning of dialysis can be helpful for identifying those patients at higher risk of experiencing an IDH episode, and consequently nephrologists might consider reducing/minimizing ultrafiltration. Conversely, the identification of <5 B-lines at the end of a dialysis session complicated by an IDH episode is highly suggestive of dehydration, probably due to an overestimation of the dry weight, and consequently nephrologists might consider increasing the dry weight.

**Table 1 diagnostics-12-02990-t001:** Clinical characteristics of the study population: patients on hemodialysis who underwent multiparametric assessment.

	All	No Intradialytic Hypotension Group	Intradialytic Hypotension Group	*p*-Value
Number of patients	107	74	33	
Males, n (%)	68 (63.6)	48 (64.9)	20 (60.6)	0.672
Age [years, median (IQR)]	69.1 (58.2–81.3)	68.2 (58.1–79.6)	75.8 (59.1–82.1)	0.324
**Comorbidities:**				
Diabetes, n (%)	28 (26.7%)	18 (24.3)	10 (30.3)	0.516
Hypertension, n (%)	99 (92.5)	70 (94.6)	29 (87.9)	0.124
Dialysis vintage [years, median (IQR)]	1.7 (0.2–3.5)	1.6 (0.5–3.7)	1.8 (0.2–4.9)	0.589
Oligoanuria, n (%)	60 (56.1)	41 (55.4)	19 (57.6)	0.835
LVH, n (%)	87 (81.3)	63 (85.1)	24 (72.7)	0.128
NYHA ≥ 3, n (%)	26 (24.3)	13 (17.6)	13 (39.4)	**0.015**
Anemia, n (%)	43 (40.1)	29 (39.2)	14 (42.4)	0.753
Hypoalbuminemia, n (%)	43 (40.1)	27 (36.5)	16 (48.5)	0.242
**Pre-dialysis fluid status assessment:**				
SBP before dialysis [mmHg, median (IQR)]	135 (120–147)	137 (130–150)	126 (110–138)	**0.003**
Peripheral oedema-pulmonary crackles, n (%)	42 (39.3)	31 (41.9)	11 (33.3)	0.632
Lung congestion at chest X-ray, n (%)	52 (47.7)	43 (58.1)	9 (27.3)	**0.006**
NT-proBNP [pg/mL, median (IQR)]	8896 (3545–34,500)	22,249 (3888–63,809.8)	6306 (2899–20,310.5)	**0.019**
Interdialytic weight gain [kg, median (IQR)]	2.3 (1.3–3.5)	2.4 (1.5–3.5)	1.8 (1.0–2.9)	0.2
Interdialytic weight gain [%, median (IQR)]	3.3 (1.9–5.1)	3.6 (2.0–5.3)	2.6 (1.5–4.2)	0.117
UF rate [mL/Kg/hr, median (IQR)]	10.5 (7.3–12.7)	10.5 (6.9–13.5)	10.5 (7.4–11.4)	0.684
B-lines before dialysis [n, median (IQR)]	15.0 (6.0–35.0)	18 (6.9–13.5)	7.0 (3.0–15.5)	**<0.001**
B-lines ≤ 5 before dialysis, n (%)	26 (24.3)	12 (16.2)	14 (42.4)	**0.004**
B-lines ≥ 15 before dialysis, n (%)	55 (51.4)	45 (60.8)	10 (30.3)	**0.004**
**Fluid status assessment all along the dialysis session:**				
B-lines after dialysis [n, median (IQR)]	3.0 (1.0–17.0)	4.5 (1.0–22.8)	1.0 (0–3.0)	**0.006**
% slope in RBV during first hour of dialysis [n, median (IQR)]	−5.5 (−2.8–−8.5)	−4.6 (−4.2–−6.7)	−7.6 (−4.2–−10)	**<0.001**
**Follow-up**				
12-month mortality, n (%)	31/92 (33.7)	17/60 (28.3)	14/32 (43.8)	0.14

Legend: RBV, relative blood volume; LVH, left ventricular hypertrophy; NYHA, New York Heart Association; IQR, interquartile range; Hb, hemoglobin; UF, ultrafiltration; SBP, systolic blood pressure.

**Table 2 diagnostics-12-02990-t002:** Clinical characteristics of patients with heart failure (defined as NYHA class ≥3) on hemodialysis who underwent multiparametric assessment.

	All	No intradialytic Hypotension Group	Intradialytic Hypotension Group	*p*-Value
Number of patients	26	13	13	
Males, n (%)	20 (77)	10 (77)	10 (77)	1.0
Age [years, median (IQR)]	76.6 (71.7–84.2)	73.6 (66.5–84.9)	79.6 (76.0–83.9)	0.41
**Different methods for fluid status assessment before dialysis:**				
SBP before dialysis [mmHg, median (IQR)]	128 (114–138)	130 (114–138)	126 (114–138)	0.84
Peripheral oedema-pulmonary crackles, n (%)	15 (57.7)	9 (69.2)	6 (46.2)	0.43
Lung congestion at chest X-ray, n (%)	18 (69.2)	12 (92.3)	6 (46.2)	**0.03**
NT-proBNP [pg/mL, median (IQR)]	49,895 (11,681–82,920)	76,293 (48,141–84,818)	21,097 (6279–23,784)	**<0.001**
B-lines before dialysis [n, median (IQR)]	28.6 (12.8–47.5)	41.8 (37–51)	15.4 (3–18)	**0.001**
**Different methods for fluid status assessment during the dialysis session:**				
% slope in RBV during first hour of dialysis [n, median (IQR)]	−5.1 (−7.7–−3.0)	−4.1 (−5.5–−2.8)	−6.2 (−9.5–−3.0)	0.24
B-lines after dialysis [n, median (IQR)]	9 (3–18)	24 (10–28)	3 (0–6)	**<0.001**

Legend: RBV, relative blood volume; NYHA, New York Heart Association; IQR, interquartile range; SBP, systolic blood pressure.

**Table 3 diagnostics-12-02990-t003:** Multivariable analysis of different risk factors predicting an intradialytic hypotension episode.

	Model 1				Model 2			
	OR	95% CI		*p*	OR	95% CI		*p*
Age (years)	0.989	0.950	1.029	0.579	-	-	-	-
Gender (M)	0.666	0.211	2.100	0.487	-	-	-	-
NYHA class	2.334	0.989	5.509	0.053	2.06	1.027	4.132	**0.042**
Previous cardiovascular events (yes/no)	0.809	0.199	3.298	0.768	-	-	-	-
Residual diuresis (mL)	0.999	0.998	1.00	0.192	-	-	-	-
SBP before dialysis (mmHg)	0.964	0.933	0.996	**0.030**	0.965	0.938	0.994	**0.017**
Delta weight gain/dry weight (%)	1.149	0.781	1.689	0.480	-	-	-	-
UF rate (mL/kg/h)	1.018	0.832	1.246	0.863	-	-	-	-
Peripheral oedema-pulmonary crackles (yes/no)	4.476	0.980	21.369	0.53	-	-	-	-
Hypoalbuminemia (yes/no)	0.319	0.091	1.122	0.075	0.471	0.164	1.352	0.162
B-lines before dialysis (n)	0.877	0.817	0.942	**<0.001**	0.920	0.881	0.962	**<0.001**
Hb (g/dL)	1.364	0.397	4.69	0.622	-	-	-	-

Model 1: all the variables included in the model. Model 2: NYHA class, SBP before dialysis, hypoalbuminemia, B-lines before dialysis included. Legend: NYHA, New York Heart Association; IQR, interquartile range; Hb, hemoglobin; UF, ultrafiltration per hour; SBP, systolic blood pressure.

## Data Availability

The datasets used and/or analyzed during the current study are available from the corresponding author on reasonable request.

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
