# Peer review of "Pre-Dialysis B-Line Quantification at Lung Ultrasound Is a Useful Method for Evaluating the Dry Weight and Predicting the Risk of Intradialytic Hypotension"

_diagnostics, 2022, doi:10.3390/diagnostics12122990_

Round 1
Reviewer 1 Report
Really good job.
Mimtiparametric evaluation of fluid status is very well described, and LUS is properly applied in this.
I don’t use the number of B-lines to identify the fluid status or interstitial diseas, because I prefer LUS-score ad a semi-quantitive/qualitative (to define interstitial ecographic disease) score, but in literature also your method is well validated.
Author Response
Really good job.
Multiparametric evaluation of fluid status is very well described, and LUS is properly applied in this.
I don’t use the number of B-lines to identify the fluid status or interstitial disease, because I prefer LUS-score ad a semi-quantitative/qualitative (to define interstitial ecographic disease) score, but in literature also your method is well validated.
Thank you very much for your kind comments.
Reviewer 2 Report
Dear author, the paper:” Pre-dialysis B-line quantification at lung ultrasound is a useful 2 method for evaluating the dry weight and predicting the risk of 3 intradialytic hypotension” has a very interesting topic in practical hemodialysis to prevent the IDH episodes.
1) The author described the study as a prospective study. However, it is not clear from the document whether it is a transversal or prospective study. The author seems to have evaluated a hemodialysis session without a follow-up. If true, the study is a transversal and not a prospective study.
2) How did the author classify IDH patients? Were they patients who had an IDH episode when the LUS was done? Alternatively, the author ranked the patient’s baseline as IDH and non-IDH and then rated the LUS? In the latter case, we need to know the method that the author used to classify patients in IDH or non-IDH.
3) The main objective of the study is unclear. The author states in the introduction: “This prospective cohort study aimed to evaluate the relationship between IDH episodes and different parameters to identify the most important predictive factors of IDH". Is it the main purpose or is it to find a method to evaluate the dry weight of patients? Because from the result I understood that, the IDH episodes were linked to a wrong evaluation of dry weight.
4) It is important to know if all patients (IDH and non-IDH) target the estimated dry weight during the haemodialysis session in the study. Because from the table I cannot understand if the non-IDH population did not target the estimated dry weight. This could be important information to understand if without the LUS evaluation all patients were considered a target weight after hemodialysis and only with the LUS evaluation, we were able to understand that it is not true.
5) It is useful to know how long the LUS evaluation lasted.
6) It is useful to know the type of vascular access of patients, KT/V, HD prescription for sodium, temperature, Qb and time
Author Response
1) The author described the study as a prospective study. However, it is not clear from the document whether it is a transversal or prospective study. The author seems to have evaluated a hemodialysis session without a follow-up. If true, the study is a transversal and not a prospective study.
Thank you for your comment. We collected several pieces of information, both in the short-term and long-term follow-up. For example, we included information about long-term mortality. We thus define our work as a prospective one.
2) How did the author classify IDH patients? Were they patients who had an IDH episode when the LUS was done? Alternatively, the author ranked the patient’s baseline as IDH and non-IDH and then rated the LUS? In the latter case, we need to know the method that the author used to classify patients in IDH or non-IDH.
We thank the reviewer for this comment, which allows us to better clarify the study design. We thus expanded the Methods section accordingly. Patients were divided into a hypotension group (IDH) and non-hypotension group (no-IDH) given the achievement of the primary outcome (an IDH episode) experienced the same day as the multiparametric evaluation. At the time of patients recruitment, and so before the hemodialysis session, it was not possible to predict in which group the patient would be included, and in order not to create selection bias all patients of our Dialysis Centers were progressively recruited.
3) The main objective of the study is unclear. The author states in the introduction: “This prospective cohort study aimed to evaluate the relationship between IDH episodes and different parameters to identify the most important predictive factors of IDH". Is it the main purpose or is it to find a method to evaluate the dry weight of patients? Because from the result I understood that, the IDH episodes were linked to a wrong evaluation of dry weight.
Our study was designed mainly to understand if B-line quantification is a good method to predict those IDH events due to dehydration. We modified the sentence as: “This prospective cohort study aimed to evaluate the potential role of LUS, and in particular of B-line quantification, in predicting IDH episodes and detecting patients at greater risk of developing IDH because of their fluid status or overestimation of the dry weight. We also evaluated the relationship between IDH episodes and different parameters to help clarifying the most important predictive factors of IDH”.
4) It is important to know if all patients (IDH and non-IDH) target the estimated dry weight during the haemodialysis session in the study. Because from the table I cannot understand if the non-IDH population did not target the estimated dry weight. This could be important information to understand if without the LUS evaluation all patients were considered a target weight after hemodialysis and only with the LUS evaluation, we were able to understand that it is not true.
Again, thank you for your interesting point of view. Not all hypotension is due to hypovolemia, and consequently, several patients did not reach the dry weight. It might be caused by an excessive UF/rate, a combination of heart failure plus severe interdialytic weight gain, etc. We included the following sentence in the Discussion: “a post-dialysis B-line quantification may allow understanding if the IDH episode is due to dehydration, demonstrated by <5 B-lines, or other reasons (e.g. autonomic dysfunction, high UF rate, non-dialyzable antihypertensive drugs, hypoalbuminemia and/or anemia).”
5) It is useful to know how long the LUS evaluation lasted.
We included the following sentence: “The duration of ultrasound bedside assessments ranged from 5 to 8 min,”.
6) It is useful to know the type of vascular access of patients, KT/V, HD prescription for sodium, temperature, Qb and time
Unfortunately, we did not collect all these variables in this study and we are not able to retrieve all of them. As depicted in Table 1, the IDH and no-IDH groups displayed comparable clinical characterics.
Reviewer 3 Report
Allinovi et al. investigated 107 hemodialysis patients with parallel determination of B-Lines in Lung-ultrasound pre- and post dialysis, clinical hydration status, NT-proBNP measurement and predialysis X-ray. The authors retrospectively divided the patients in a group with “symptomatic” intradialytic hypotension (IDH) and one without. Low B-Line numbers were associated with IDH and high numbers were associated with fluid overload.
- How many patients had a systolic blood pressure drop of more than 20 mmHg and how many patients had a symptomatic drop?
- Table 1: UF-rate should be no higher than 10 ml/Kg BW per hour. In this study the mean UF rate was higher. This could account for the relatively high incidence of IDH
- How much time did investigating 28 positions with lung ultrasound cost. Although this is an interesting method it might be too time consuming.
- Patients with severe heart failure (BNP high) were more prone to develop IDH (B-lines low) and B-line number (high) correlated positively with BNP values (high). How do these two observations fit?
- Page 6 line 208: Increasing B-Lines was associated with a decreased likelihood of IDH. Isn´t this the other way round? Additionally a 0.946 times higher odds ratio with a single increase in a B-line should be better named lesser odd because the odds ratio of 0,946 suggests a 5.4% reduced risk.
- Page 8 line 267 ff: Mortality was not different whether the patients had IDH or whether they had decreased (or increased) numbers of B-Lines. In the introduction the authors stated that IDH was associated with an increased likelihood of death. How is this explained? Is it due to the low number of patients. What percentage of patients with IDH and what percentage of patients without IDH died?
Minor
- What was the prospective part in this observational study? Grouping as IDH yes or no was perfomed retrospectively and the determination of the best cut-off value for B-Lines was also done retrospectively.
- Was X-ray performed predialysis only?
- Page 8 line 269: It should be … neither … nor and not nor… neither.
- Page 9 line 302: What is the cutoff value for detection of IDH when using post dialysis B-line number as a marker of IDH
Author Response
Allinovi et al. investigated 107 hemodialysis patients with parallel determination of B-Lines in Lung-ultrasound pre- and post dialysis, clinical hydration status, NT-proBNP measurement and predialysis X-ray. The authors retrospectively divided the patients in a group with “symptomatic” intradialytic hypotension (IDH) and one without. Low B-Line numbers were associated with IDH and high numbers were associated with fluid overload.
- How many patients had a systolic blood pressure drop of more than 20 mmHg and how many patients had a symptomatic drop?
92% of patients experienced a systolic blood pressure drop of more than 20 mmHg, and 68% of patients experienced an intra-dialyitic symptomatic drop.
- Table 1: UF-rate should be no higher than 10 ml/Kg BW per hour. In this study the mean UF rate was higher. This could account for the relatively high incidence of IDH
We totally agree with the reviewer. Our center frequently adopts a high UF/rate, but our average UF/rate is not far from the values described in several published papers.
- How much time did investigating 28 positions with lung ultrasound cost. Although this is an interesting method it might be too time consuming.
We have included the following sentence into the results: "The duration of ultrasound bedside assessments ranged from 5 to 8 min, and female gender and obesity were generally associated with longer execution times."
- Patients with severe heart failure (BNP high) were more prone to develop IDH (B-lines low) and B-line number (high) correlated positively with BNP values (high). How do these two observations fit?
Thank you for your comment. We added the following sentence in the Discussion section to clarify this aspect: “Each method (physical examination, B-line number, blood pressure measurement, NTproBNP) suffers from several shortcomings, and consequently we should adopt different methods or different thresholds for each method in presence of comorbidities associated with autonomic dysfunction. For example, in patients with severe heart failure, the loss of compensation from increased contractility predisposes to the development of IDH even with >8 pre-dialysis B-lines or with physical signs of fluid overload.”
- Page 6 line 208: Increasing B-Lines was associated with a decreased likelihood of IDH. Isn´t this the other way round? Additionally a 0.946 times higher odds ratio with a single increase in a B-line should be better named lesser odd because the odds ratio of 0,946 suggests a 5.4% reduced risk.
We totally agree with you. Sorry for the mistake. We changed the sentence accordingly: “A single increase in B-lines number confers 0,946 times lower odds to experience an IDH episode (95% CI 0,913 - 0,979, p=0,003), thus indicating that a higher B-line number was associated with a decreased likelihood of IDH.”
- Page 8 line 267 ff: Mortality was not different whether the patients had IDH or whether they had decreased (or increased) numbers of B-Lines. In the introduction the authors stated that IDH was associated with an increased likelihood of death. How is this explained? Is it due to the low number of patients. What percentage of patients with IDH and what percentage of patients without IDH died?
12-month mortality did not differ between IDH and non-IDH patients (43.8% vs 28.3%, p=0,136) but we can appreciate a trend. We believe that it did not reach statistical significance owing to the relatively low number of patients.
Minor
- What was the prospective part in this observational study? Grouping as IDH yes or no was perfomed retrospectively and the determination of the best cut-off value for B-Lines was also done retrospectively.
Thank you for your comment. We collected several pieces of information, both in the short-term and long-term follow-up. For example, we included information about long-term mortality. We defined the design of the study before patient recruitment. We started with a hypothesis and with a specific approach before patient recruitment. We followed our patients for 12-18 months to understand a potential correlation between IDH and mortality. We thus define our work as a prospective one.
- Was X-ray performed predialysis only?
X-ray assessment occurred before the dialysis session. We included the following sentence: “Each patient underwent a chest X-ray before the dialysis session.” We only rarely performed chest X-ray after dialysis.
- Page 8 line 269: It should be … neither … nor and not nor… neither.
We modified accordingly.
- Page 9 line 302: What is the cutoff value for detection of IDH when using post dialysis B-line number as a marker of IDH
Thank you for this suggestion. With regard to post-dyalisis B-lines number, a lower number of B-lines was detectable in patients who experienced IDH (OR 0,895, 95% CI 0,7830 - 0,9684). ROC analysis retrieved an AUC at an acceptable level of discrimination (AUC=0,734, 95% IC 0,5843 - 0,8841), with an optimal threshold at 5 B-lines. We added a new ROC curve (Figure 1C) and modified the Results section accordingly.
Reviewer 4 Report
Thank you for the opportunity to review quantification of B-lines by lung ultrasound by Allinovi and others.
1. This is a prospective observational study that conducted a SINGLE multi-parametric evaluation (pre and post dialysis) on select 107 patients (not consecutive patients – page 9, 345-347) during the study period, at a SINGLE tertiary care centre.
2. Page 3, lines 107-109; claims patients with multiple B lines regardless of fluid status were excluded from study, then authors appear to include patients with advanced heart failure, where the up to 20 B-lines were felt to be acceptable, more than over 15 B-lines that suggests fluid overload in other patients. How come these patients were included in the study?
3. The study should have compared with hemocontrol or blood volume monitoring methods that are commonly applied in dialysis units to adjust the ultrafiltration rates to avoid IHD.
4. Intradialytic hypotension (IDH), as authors state (page 2, Lines 46-48) is mainly due to imbalance between ultrafiltration rate and normal compensatory mechanisms – venous refilling. The dry weigh contributes to some extent but most important determinants of IDH are the interdialytic weight gain, and impaired compensatory mechanisms, especially reduced venous refilling because of hypoalbuminemia, and in such circumstances.
5. Furthermore, LUS is not discriminatory in patients with heart failure, where over 20 B-lines, compared to <8 B lines, was associated with risk of IDH. However, as noted in table 1, there were 26 patients with Heart failure, NYHA class >3 or higher but the definition applied to assess these patients was the same as in patients without heart failure, when less than 20 B-lines in patients with heart failure was associated with IDH, compared to <8 B-lines with patients without CHF.
6. As only single LUS was performed (pre-and post dialysis), It is not clear how authors can assess the effect of an “increase in single B-line’ as claimed on page 6, lines 219-220).
7. Does the size of US probe affect the number of counted B-lines in the scanned area?
8. It would be best if authors could have included a video of B-lines, in few fields to assess the technique accuracy.
9. The systolic blood pressure before dialysis was as predictive of B-lines (Table 1).
Unfortunately, there are many issues with methodology that by itself is contributing to significant bias in this study and conclusion of advocating use of predialysis LUS is not supported without significant bias, and interestingly authors do acknowledge this on page 9, lines 343-345.
Author Response
This is a prospective observational study that conducted a SINGLE multi-parametric evaluation (pre and post dialysis) on select 107 patients (not consecutive patients – page 9, 345-347) during the study period, at a SINGLE tertiary care centre.
- Page 3, lines 107-109; claims patients with multiple B lines regardless of fluid status were excluded from study, then authors appear to include patients with advanced heart failure, where the up to 20 B-lines were felt to be acceptable, more than over 15 B-lines that suggests fluid overload in other patients. How come these patients were included in the study?
Thank you for your interesting question. We tried to clarify our point-of-view: “…coexistent lung fibrosis or interstitial lung disease which are diseases that appear as multiple B-lines at LUS regardless of the fluid status”.
- The study should have compared with hemocontrol or blood volume monitoring methods that are commonly applied in dialysis units to adjust the ultrafiltration rates to avoid IHD.
Thank you for this suggestion, we believe this can allow us to strengthen our study. We have included in the Methods section the following sentences: “Blood volume monitoring is routinely performed in all patients undergoing HD in our unit. Nowadays, most manufacturers have incorporated a relative blood volume (RBV) monitor in their dialysis apparatus in order to monitor the RBV slope. In our cohort, the blood volume change during dialysis was monitored using HEMOcontrol BV sensor (Baxter Healthcare Corp., IL, USA) or Blood Volume Management (BVM®) (Fresenius Medical Care, Concord, CA 94520, USA). During ultrafiltration, as fluid is removed from a hemodialysis patient’s vascular space, the RBV slope (%) continuously correlates with the increase of hematocrit or total proteins. Trained dialysis nurses recorded BV reduction every hour during the session.”. Moreover, we have inserted in table 1 the “% slope in RBV during first hour of dialysis [n, median (IQR)]” as a variable.
We decided then to test the performance of BVM in predicting IDH, and we compared it with the number B-lines and commonly used clinical means. Results are displayed in a new Figure (Fig. 2) and in the section 3.8 of the Results.
- Intradialytic hypotension (IDH), as authors state (page 2, Lines 46-48) is mainly due to imbalance between ultrafiltration rate and normal compensatory mechanisms – venous refilling. The dry weigh contributes to some extent but most important determinants of IDH are the interdialytic weight gain, and impaired compensatory mechanisms, especially reduced venous refilling because of hypoalbuminemia, and in such circumstances.
We totally agree with the reviewer, and we included the following sentence: “Although two of the most important determinants of IDH are the interdialytic weight gain and impaired compensatory mechanisms (especially reduced venous refilling because of hypoalbuminemia), neither the interdialytic weight gain nor the hypoalbuminemia resulted significantly different in the IDH group (Table 1).”
- Furthermore, LUS is not discriminatory in patients with heart failure, where over 20 B-lines, compared to <8 B lines, was associated with risk of IDH. However, as noted in table 1, there were 26 patients with Heart failure, NYHA class >3 or higher but the definition applied to assess these patients was the same as in patients without heart failure, when less than 20 B-lines in patients with heart failure was associated with IDH, compared to <8 B-lines with patients without CHF.
We included the new Table 2 in order to clarify this topic. We also included a new paragraph into the discussion.
- As only single LUS was performed (pre-and post dialysis), It is not clear how authors can assess the effect of an “increase in single B-line’ as claimed on page 6, lines 219-220).
We clarified this sentence: “A single increase in B-lines number confers 0,946 times lower odds to experience an IDH episode (95% CI 0,913 - 0,979, p=0,003), thus indicating that an higher B-line number was associated with a decreased likelihood of IDH.”
- Does the size of US probe affect the number of counted B-lines in the scanned area?
We only adopted a 6-18 MHz linear probe.
- It would be best if authors could have included a video of B-lines, in few fields to assess the technique accuracy.
Thank for your suggestion. Scientific literature and the internet are rich in videos explaining how to perform a B-line quantification using the standardized 28-position B-line score. This is a widely accepted and validated lung ultrasound technique with more than 10 years of experience. So, we decided not to include a video in this work.
- The systolic blood pressure before dialysis was as predictive of B-lines (Table 1).
Table 1 is focused on IDH. Consequently, we should say that systolic blood pressure before dialysis was as predictive of IDH episode.
Unfortunately, there are many issues with methodology that by itself is contributing to significant bias in this study and conclusion of advocating use of predialysis LUS is not supported without significant bias, and interestingly authors do acknowledge this on page 9, lines 343-345.
Thank you for your comment. We strongly agree with you.
Round 2
Reviewer 2 Report
Dear authors, thank you very much for your answer. From my point of view i don't have other review
Author Response
Thank you very much for your acknowledgment.
Reviewer 4 Report
Much improved.
Authors may consider adding that based on this study, in patients with <5 predialysis B-lines on LUS, providers should be careful with UFR rates, and more vigilant about blood volume changes during dialysis, though well designed prospective studies are required to validate this observation.
Author Response
Thank you for your acknowledgment. We do agree with you and we added the sentence you suggested in the Discussion section, line 376 and 388, and modified the Conclusions accordingly.